# Mango Leaf Disease Recognition and Classification Using Novel Segmentation and Vein Pattern Technique

**Rabia Saleem** [1], **Jamal Hussain Shah** [1,*], **Muhammad Sharif** [1], **Mussarat Yasmin** [1], **Hwan-Seung Yong** [2] and **Jaehyuk Cha** [3]

1. Department of Computer Science, COMSATS University Islamabad, Wah Campus, Islamabad 45550, Pakistan; rabia_278@hotmail.com (R.S.); sharif@ciitwah.edu.pk (M.S.); mussaratabdullah@gmail.com (M.Y.)
2. Department of Computer Science & Engineering, Ewha Womans University, Seoul 120-750, Korea; hsyong@ewha.ac.kr
3. Department of computer Science, Hanyang University, Seoul 04763, Korea; chajh@hanyang.ac.kr
* Correspondence: jhshah@ciitwah.edu.pk

**Abstract:** Mango fruit is in high demand. So, the timely control of mango plant diseases is necessary to gain high returns. Automated recognition of mango plant leaf diseases is still a challenge as manual disease detection is not a feasible choice in this computerized era due to its high cost and the non-availability of mango experts and the variations in the symptoms. Amongst all the challenges, the segmentation of diseased parts is a big issue, being the pre-requisite for correct recognition and identification. For this purpose, a novel segmentation approach is proposed in this study to segment the diseased part by considering the vein pattern of the leaf. This leaf vein-seg approach segments the vein pattern of the leaf. Afterward, features are extracted and fused using canonical correlation analysis (CCA)-based fusion. As a final identification step, a cubic support vector machine (SVM) is implemented to validate the results. The highest accuracy achieved by this proposed model is 95.5%, which proves that the proposed model is very helpful to mango plant growers for the timely recognition and identification of diseases.

**Keywords:** mango leaf; CCA; vein pattern; leaf disease; cubic SVM



## 1. Introduction

Countries dependent upon agriculture are facing a terrible threat and great loss due to plant diseases, which cause a decline in the quality and quantity of fruits and yields [1]. Pakistan is among those countries where a large amount of its income is earned by importing and producing a variety of crops, vegetables, and fruits cultivated in different areas of the country [2]. Therefore, it is necessary to identify diseased plants by implementing computer vision and image processing techniques [3,4]. Recently, deep learning (DL) techniques, specifically, convolutional neural networks (CNNs), have achieved extraordinary results in many applications, including the classification of plant diseases [5,6]. The mango is a highly popular fruit and is available in summer [7]. It is important in the agricultural industry of Pakistan due to its huge production volume. Several approaches to the detection and identification of mango plant leaf diseases have been proposed in the literature. Although a large number of diseases affect mango orchards, only some of them are causing great loss to the economy of the country. A few of the more common diseases [8] include powdery mildew, sooty mold, anthracnose and apical necrosis, as shown in Figure 1. In the current era, computer scientists are aiming to devise computer-based solutions to identify the diseases in their initial phase. This will aid farmers to safeguard the crop until it is harvested, resulting in the reduction in economic loss [9]. According to agricultural experts, naked eye observation is the traditional method used to recognize plant diseases. This is very expensive and time-consuming as it requires continuous monitoring [3,10]. Hence, it is almost impossible to accurately recognize the diseases of plant at an initial

stage. Unfortunately, very few techniques that address the diseases of the mango, also called the king of fruits, have been reported before now [11,12] due to the complicated and complex structure and pattern of the plant. Hence, there is a need for efficient and robust techniques to identify mango diseases automatically, accurately, and efficiently [4,13–15]. For this purpose, the images used as a baseline can be captured by digital and mobile cameras [16,17].

Machine learning (ML) plays an important role in the identification of diseases [16,18]. ML is a sub-branch of artificial intelligence (AI) [19]. It enables computer-based systems to provide accurate and precise results. Real-world objects are the main objects to inspire ML techniques [20]. A few computerized techniques, for instance, segmentation by K-means and classification using SVM to identify the diseased area, are reported in [21].

Hence, as per the available and above discussion, there is a strong need for the automatic detection, identification, and classification the diseases of the mango plant. Keeping this in mind, this article covers the following issues: data augmentation to increase the dataset; tracking the color, size, and texture of the diseased part of leaf; handling background variation and the diseased part; the proper segmentation of the unhealthy part of the leaf; and, at a later stage, robust feature extraction and fusion to classify the disease. The following key contributions were performed to achieve this task:

1.  Image resizing and augmentation in order to set query images.
2.  A method for the segmentation of the diseased part.
3.  Fusion of color and LBP features by performing canonical correlation analysis (CCA).
4.  Using classifiers of ten different types to perform identification and recognition.

## 2. Literature Review

In current times, various types of techniques and methods have been established for the detection of plant leaf diseases. These are generally characterized into disease finding or disease detection methods and disease sorting or disease classification methods [22]. Many techniques use segmentation, feature fusion, and image classification implemented on cotton, strawberry, mango, tomato, rice, sugarcane, and citrus. Similarly, these methods are appropriate for leaf, flower, and fruit diseases because they use proper segmentation, feature extraction, and classification. To make a computer-based system work efficiently, ML techniques are generally used to enhance the visualization of the disease symptoms and to segment the diseased part for classification purposes. Mango fruit is important in the agricultural sector due to its massive production volume in Pakistan. Therefore, several approaches for the detection and identification of mango plant leaf diseases have been proposed to prevent a loss of harvest.

Iqbal et al. [23] considered segmentation, recognition, and identification techniques. They found that almost all the techniques are in the initial stage. Moreover, they discussed almost all existing methods along with their advantages, limitations, challenges, and models of ML (image processing) for the recognition and identification of diseases.

Shin et al. [24], in their study on powdery mildew of strawberry leaves, achieved an accuracy of 94.34% by combining an artificial neural network (ANN) and speeded-up robust features (SURF). However, by using an SVM and GLCM, their highest classification accuracy was 88.98%. They used HOG, SURF, GLCM, and two supervised ML methods (ANN and SVM).

Pane et al. [25] adopted an ML technique using a wavelength between 403 and 446nm for the detection of the blue color. They precisely distinguished unhealthy and healthy leaves of wild rocket, also called salad leaves. Bhatia et al. [26] used the Friedman test to rank multiple classifiers and post hoc analysis was also performed using the Nemenyi test. In their study, they found the MGSVM to be the superior classifier with an accuracy of 94.74%. Lin et al. [17] proved their results to be 3.15% more accurate than the traditional method used on pumpkin leaves. They used PCA in order to obtain 97.3% accurate results.

Shah [27] extracted the color features to detect diseases on cotton leaves. Kahlout et al. [28] developed an expert system for the detection of diseases including powdery

mildew and sooty mold on all members of the citrus family. Sharif et al. [29] recommended a computerized system to segment and classify the diseases of citrus plants. In the first part of their suggested system, they used an optimized weight technique to recognize unhealthy parts of the leaf. Secondly, color, geometric, and texture descriptors were combined. Lastly, the best features were nominated by a hybrid feature selection technique consisting of the PCA approach called entropy and they obtained 90% accuracy. Udayet et al. [30] proposed a method to classify anthracnose (diseased) leaves of mango plants. They used a multiple layer convolutional neural network (CNN) for this task. The method was applied to 1070 images collected by their own cameras and gadgets. Consequently, the classification accuracy was raised to 97.13%. Kestur et al. [31] presented a segmentation method in 2019 based on deep learning called Mango net. The results using this method are 73.6% accurate. Arivazhagan et al. [11] used a CNN and showed 96.6% accurate outcomes. No preprocessing or feature extraction was performed in this proposed technique. Srunitha [32] detected unhealthy regions for mango diseases, including red rust, anthracnose, powdery mildew, and sooty mold.

Sooty mold and powdery mildew both create a layer on the leaf by disturbing its vein pattern, as the vein is a vital part of the plant. No techniques have been proposed to make this specific diagnosis, which is a major weakness in the available literature. So, a novel segmentation technique was constructed in this work that segments the diseased part by considering the vein pattern of mango leaves. After segmentation, two features (color and texture) are extracted and used for fusion and classification purposes. We used a self-collected dataset to accomplish this task. The dataset was collected using mobile cameras and digital gadgets. Details about this work are given in Section 3. A concise and precise description of the experimental results and their interpretation is given in Section 4, while Section 5 presents the conclusion drawn from those results.

## 3. Material and Method

The primary step of this work was the preparation of a dataset. The dataset used for this work was a collection of self-collected images that were captured using different types of image capturing gadgets. These images were collected from different mango growing regions in Pakistan, including Multan, Lahore, and Faisalabad, in the form of RGB images. The collected images were resized to $256 \times 256$ after being annotated by expert pathologists, as they showed dissimilar sizes. Figure 1 presents the workflow of the proposed technique, which comprises the following steps: (1) the preprocessing of images consisting of data augmentation followed by image resizing; (2) the use of the proposed model to segment the images obtained as a result of the resizing operation (codebook); (3) color and texture (LBP) feature extraction. Finally, the images were classified by using 10 different types of classifiers.

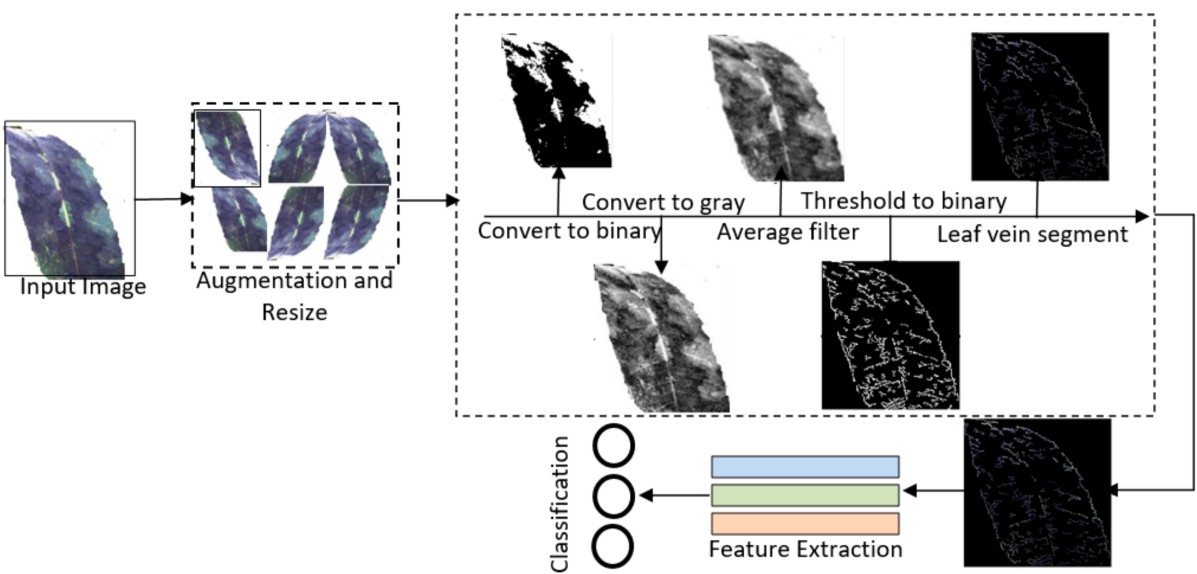

**Figure 1.** Framework of the proposed computerized system.

### 3.1. Preprocessing

The purpose of preprocessing is to improve the segmentation and classification accuracy by enhancing the quality of the image. The detailed sketch of each phase implemented for this purpose is as follows:

Resizing and Data Augmentation

A total of 29 images of healthy and unhealthy mango plant leaves were collected. Some of the images (2 out of 29) were distorted upon applying the resize operation. The distorted images were discarded, and the remaining 27 images were augmented by flipping and rotating [33] them horizontally, vertically, and both horizontally and vertically, as well as by using power-law transformations with gamma = 0.5 and c = 1, as shown in Figure 2. As such, 135 images were made available for tuning the proposed algorithm, as shown in Table 1.

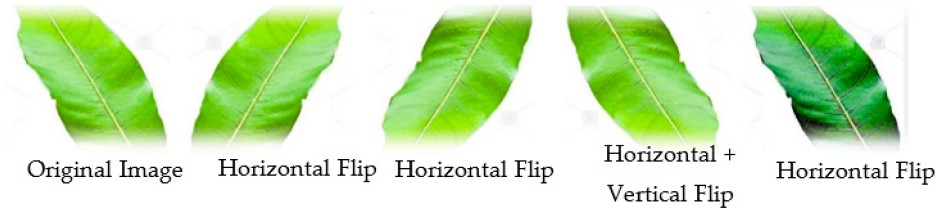

| Original Image | Horizontal Flip | Horizontal Flip | Horizontal + Vertical Flip | Horizontal Flip |

**Figure 2.** Data augmentation steps.

**Table 1.** Distribution of the data.

| Sooty Mold | Powdery Mildew | Healthy | Total |
|:---:|:---:|:---:|:---:|
| 45 | 45 | 45 | 135 |

An equal image size of 256 × 256 was utilized for the current study. From the whole dataset, only diseased images were used for segmentation, while the other 45 images of healthy leaves were used for classification.

### 3.2. Proposed Leaf Vein-Seg Architecture

The second step after preprocessing is the proposed leaf vein-seg architecture. The powder-like, purplish-white fungi growing mainly on the leaves that makes the plants dry and brown is called powdery mildew [34]. The honey-dew-like insect secretions that form a brownish layer on the leaf of mango plants are called sooty mold.

The proposed architecture is a stepwise process that extracts the veins of the leaf. The extracted veins are helpful in further processing and in the classification of diseases of the mango plant. First, the RGB input image is converted to a binary image. The binary image is then converted to a gray-scale image to extract a single channel from the image. CLAHE is applied to it, which improves the quality of the image by improving its contrast. CLAHE operates on small regions of an image called tiles and takes care of the over amplification of contrast in an image [35]. Bilinear interpolation is used to remove artificial boundaries by combining the neighboring tiles.

An average filter of $9 \times 9$ is then applied to the output gray-scale image to exclude its background. A mean or average filter smooths the image by reducing the intensity variation among the neighbor pixels. This filter works by replacing the original value of a pixel with the average value of its own and neighboring pixels. It moves pixel-by-pixel through the whole image. An average $9 \times 9$ filter is shown in matrix form below. Mathematically, it can be represented by Equation (1).

$$I_{new(x,y)} = \sum_{j=-1}^{1} \sum_{i=-1}^{1} 1 \times I_{old}(x+i, y+j) \tag{1}$$

The output is then normalized to make the image pixel values between 0 and 255 as given in Equation (2).

$$I_{new(x,y)}^{normalized} = \frac{1}{\sum_{j=-1}^{1} \sum_{i=-1}^{1} 1} \sum_{j=-1}^{1} \sum_{i=-1}^{1} 1 \times I_{old}(x+i, y+j) \tag{2}$$

The edges of the image are detected, and noise is removed by sharpening the image. In the next step, the difference is calculated from the images obtained in the previous two steps: the gray-scale images and the images obtained after the application of the average filer along with its normalization, as the images should be the same size. This step is performed to compare the images and correct uneven luminance. Its mathematical representation is shown in Equation (3):

$$D(x,y) = I_1(x,y) - I_{new(x,y)}^{normalized} \tag{3}$$

where the output of the gray-scale image is $I_1(x,y)$ and $I_{new(x,y)}^{normalized}$ is the output after the application of the average filter and its normalization. The threshold is then applied to the obtained images as $D(x,y)$. A method was designed to perform this task. The global image threshold is computed by setting a threshold level to $D(x,y)$, obtained in Equation (3). The computed global threshold level is used to convert the intensity image to a binary image. Standardized intensity values lie between 0 and 1. The histogram is segmented into two parts to normalize the image by using a starting threshold value that should be half of the maximum dynamic range.

$$H = 2B - 1 \tag{4}$$

Foreground and background values are computed by using the sample mean values. Mean values associated with the foreground are (mf, 0), and the gray values associated with the background are (mb, 0). In this way, threshold value 1 is computed and this process is repeated until the threshold value does not change any more. The image obtained is converted to a binary image. The next step is to take the complement of binary image D. All zeros became one and all ones became zero. Mathematically, it can be represented in Equation (5).

$$D = D' \tag{5}$$

Another method was designed to obtain the final segmented output in the form of edges and veins that is used to detect the diseased leaf of the mango plant. It takes as input the binary image obtained in Equation (5) to produce a mask of indicated colors for the output image while using a $1 \times 3$ size vector with values between 0 and 1 for the color. The mask must be represented in a logical two-dimensional matrix where [0 0 0] shows white and [1 1 1] shows black. The output of this step is a segmented RGB image. The processed images, including both diseased and healthy leaves of a mango plant, are presented in Figure 3.

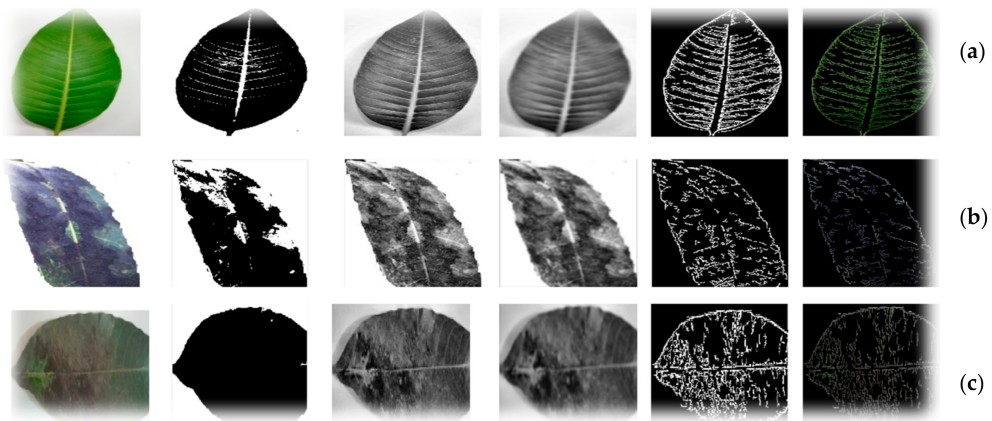

**Figure 3.** Stepwise output of leaves: (**a**) healthy, (**b**) powdery mildew, and (**c**) sooty mold.

### 3.3. Features Extraction and Fusion

This is a helpful phase of an artificially intelligent, automated system based on ML. Color and texture features are helpful descriptors. Information about the color is obtained by using color features, whereas texture features provide texture analysis for the diseased leaves of the mango plant. In order to conduct the classification of diseases (powdery mildew, sooty mold) of the mango plant, color, shape, and texture features are extracted in this proposed method. The structure of the feature (color, shape, and texture) extraction is shown in Figure 4.

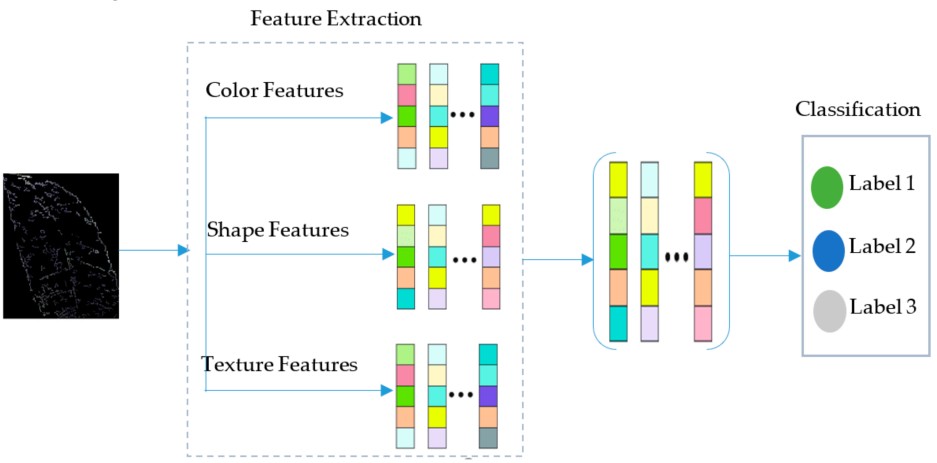

**Figure 4.** Structure of feature extraction.

The comprehensive explanation of each phase is as follows: preprocessed images are passed through the codebook and a segmented image is achieved in the training phase; the segmented image is used for feature extraction; and then color, shape, and texture features are extracted.

Color Features: These are a significant resource for the detection and recognition of diseased/unhealthy parts of mango plants, as every mango plant leaf disease has a different color and shade. In this paper, the diseases on mango plant leaves are recognized or identified by the extraction of color features. As stated earlier, each disease has its own pattern and shading, so four types of color spaces are utilized to obtain the extracted colors. Features including RGB; hue, saturation, and variance (HSV); luminance, a, b component (LAB); hue, intensity, and saturation (HIS) are obtained from the mango leaf images. Different information is obtained along every channel. Therefore, the maximum information about the defective part of the mango leaf is obtained by applying color spaces. Color features are obtained by using the six types of different statistical metrics mentioned below from Equations (6)–(11). One vector, sized 1 × 3000, is used to combine these features for each channel. The statistical metrics are calculated by the following formulae:

$$\overline{A} = \left( \sum a_i \right) / n, \tag{6}$$

$$\sigma = \frac{1}{n} \sqrt{\sum_{i=1}^{n} (a_i - a')^2} \tag{7}$$

$$V = \frac{1}{n} \sum_{i=1}^{n} (a_i - a')^2 \tag{8}$$

$$Entropy = \sum_{i=1}^{c} -p_i \log_2 p_i \tag{9}$$

$$KR = n \frac{\sum_{i=1}^{n} (A_i - A_{avg})^4}{(\sum_{i=1}^{n} (A_i - A_{avg})^2)^2} \tag{10}$$

$$Skewness = \frac{\frac{1}{n} \sum_{i=1}^{n} (a_i - a')^3}{\left( \frac{1}{n} \sum_{i=1}^{n} (a_i - a')^2 \right)^{\frac{3}{2}}} \tag{11}$$

where $\overline{A}$ denotes the mean feature, $\sigma$ denotes the standard deviation feature, $V$ represents the variance feature, *Entropy* describes the entropy, *KR* represents the kurtosis feature, and *Skewness* indicates the skewness feature.

Texture Features: Texture features alone cannot find identical images. Other features, such as color, work with texture features to segregate texture and non-texture features. To handle complications in image texture, an ancient but easy technique, local binary pattern (LBP), is implemented [36]. Hence, LBP was used in this research to extract texture features. Its description is mentioned below:

$$LBP_{A,B} = \sum_{A=0}^{A-1} F(g_A - g_c) 2^A, F(x) = \begin{cases} 1 \ if \ \text{x} \ge 0; \\ 0 \ otherwise \end{cases} \tag{12}$$

In Equation (12), the value of a pixel is denoted by $A$, the value of the radius by $B$, the neighborhood point by $g_a$, the center point by $g_c$, and the binomial factor is denoted by $2^A$. As a result, a 1 × 800 size vector was generated after extracting features from different channels and color spaces.

### 3.4. Features Fusion and Classification

Finally, a CCA-based feature reduction is applied to the extracted features. A serial-based fusion approach is used to fuse the resultant reduced vectors. A vector of dimensions $n \times 2000$ is obtained simply by concatenating the features. This is used for classification and fed into the classifiers. Ten different types of classification techniques were implemented for the analysis of classification accuracy. In the past, agricultural applications suffered from the unavailability of data due to its complex structure and collection cost, especially for mango plants because these plants are available only in select regions. The cost of labeling for data acquisition is also very high [37]. Hence, this issue encouraged us to collect data by ourselves from mango growing areas in Pakistan. We adopted 2 strategies in this research: (1) data augmentation, and (2) a segmentation technique to segment the

diseased parts and veins of mango leaves. This is a unique technique as it has not been performed in earlier agricultural applications, especially with the mango plant. As a final point, the segmented images were entered into a computer-based system for feature fusion, and then for identification. The computation was about 45 min as the segmentation of one image took almost 7.5 s. All simulations were performed on a personal computer with the following specifications: 64-bit Windows operating system with MATLAB version 2018, 32 GB RAM with an Intel® Xeon® processor and central processing unit of 2.2GHz, GPU GeForce GT×1080.

## 4. Experimental Results and Analysis

In this section, the results of the proposed algorithm are discussed in both graphic and tabular form. For validation, out of the total 135 images used, 45 images were of sooty mold, 45 images were of powdery mildew, and 45 were of healthy mango plants. As a primary step, the proposed segmentation technique was used. Second, the classification results for this segmentation technique were tabulated by implementing different standardized classifiers. The detailed results, with their descriptions, are discussed in the following section. The identification of each disease is analyzed with the images of the healthy leaves of the mango plant. Then, the classification accuracy of all diseased leaves is compared with the classification of the healthy leaves of the mango plant. The proposed technique was tested with 10 of the most ideal classifiers with 1- fold cross-validation.

### 4.1. Test 1: Powdery Mildew vs. Healthy

In this test, 45 powdery mildew and 45 healthy mango leaf images were classified. Table 2 shows that an accuracy of 96.6% was attained through cubic SVM. It proved to be the highest amongst all the other competing classifiers. Furthermore, 0.16, 0.97, 0.97, and 3.4 were the obtained values of sensitivity, specificity, AUC, and FNR, respectively. Sensitivity and specificity mathematically describe the accuracy of a test that reports the presence or absence of a condition. The confusion matrix of this test is also given in Table 3.

**Table 2.** Powdery mildew vs. healthy mango leaves.

| Methods | Sensitivity | Specificity | AUC | FNR (%) | Accuracy (%) |
|---|---|---|---|---|---|
| Linear discriminant | 0.24 | 0.89 | 0.82 | 17.8 | 82.2 |
| Linear SVM | 0.22 | 0.91 | 0.85 | 5.6 | 94.4 |
| Quadratic SVM | 0.22 | 0.91 | 0.86 | 6.7 | 93.3 |
| Cubic SVM | 0.16 | 0.97 | 0.97 | 3.4 | 96.6 |
| Fine KNN | 0.22 | 0.91 | 0.84 | 11.2 | 88.8 |
| Medium KNN | 0.18 | 0.84 | 0.86 | 16.7 | 83.3 |
| Cubic KNN | 0.20 | 0.71 | 0.84 | 18.9 | 81.1 |
| Weighted KNN | 0.18 | 0.82 | 0.86 | 11.2 | 88.8 |
| Subspace discriminant | 0.2 | 0.82 | 0.86 | 12.3 | 87.7 |
| Subspace KNN | 0.18 | 0.84 | 0.86 | 16.7 | 83.3 |

**Table 3.** Confusion matrix of powdery mildew vs. healthy mango leaves.

| Classification-Class | Classification-Class | |
|---|---|---|
| | Powdery Mildew | Healthy |
| Powdery mildew | 97.7% | <1% |
| Healthy | <1% | 95.6% |

### 4.2. Test 2: Sooty Mold vs. Healthy

In this test, 45 images of sooty mold and 45 healthy mango leaf images were classified. Table 4 shows that an accuracy of 95.5% was attained by using linear SVM. It proved to be the highest amongst all the other competing classifiers. Furthermore, 0.05, 0.95, 0.95,

and 4.5 are the obtained values of sensitivity, specificity, AUC, and FNR, respectively. The confusion matrix of this test is also given in Table 5.

**Table 4.** Sooty mold vs. healthy mango leaves.

| Methods | Sensitivity | Specificity | AUC | FNR (%) | Accuracy (%) |
|---|---|---|---|---|---|
| Linear discriminant | 0.47 | 0.74 | 0.63 | 36.7 | 63.3 |
| Linear SVM | 0.05 | 0.95 | 0.95 | 4.5 | 95.5 |
| Quadratic SVM | 0.22 | 0.91 | 0.86 | 6.7 | 93.3 |
| Cubic SVM | 0.22 | 0.91 | 0.85 | 5.6 | 94.4 |
| Fine KNN | 0.24 | 0.67 | 0.71 | 28.9 | 71.1 |
| Medium KNN | 0.2 | 0.82 | 0.86 | 12.3 | 87.7 |
| Cubic KNN | 0.23 | 0.82 | 0.86 | 11.2 | 88.8 |
| Weighted KNN | 0.20 | 0.87 | 0.93 | 12.5 | 83.3 |
| Subspace discriminant | 0.29 | 0.91 | 0.82 | 18.9 | 81.1 |
| Subspace KNN | 0.24 | 0.6 | 0.75 | 32.2 | 67.8 |

**Table 5.** Confusion matrix of sooty mold vs. healthy mango leaves.

| Classification Class | Classification Class | |
|---|---|---|
| | **Sooty Mold** | **Healthy** |
| Sooty mold | 95.5% | <1% |
| Healthy | <1% | 95.5% |

### 4.3. Test 3: Diseased vs. Healthy

This section presents the findings of the classification all unhealthy and healthy images of mango plant leaves. Table 6 shows the classification results for all the diseases obtained after feature fusion based on CCA. This test was performed on all 135 images. The accuracy of 95.5% was attained using a cubic SVM, which is the highest among all the other competing classifiers. Moreover, 0.03, 0.93, 0.99, and 4.5 were the values of the sensitivity, specificity, AUC, and FNR, respectively. The confusion matrix of this test is also given in Table 7.

**Table 6.** Diseased vs. healthy mango leaves.

| Methods | Sensitivity | Specificity | AUC | FNR (%) | Accuracy (%) |
|---|---|---|---|---|---|
| Linear discriminant | 0.13 | 0.69 | 0.78 | 27.4 | 72.6 |
| Linear SVM | 0.03 | 0.88 | 0.98 | 6.7 | 93.3 |
| Quadratic SVM | 0.14 | 0.76 | 0.88 | 25.2 | 74.8 |
| Cubic SVM | 0.03 | 0.93 | 0.99 | 4.5 | 95.5 |
| Fine KNN | 0.10 | 0.56 | 0.73 | 33.3 | 66.7 |
| Medium KNN | 0.06 | 0.83 | 0.88 | 11.2 | 88.8 |
| Cubic KNN | 0.03 | 0.89 | 0.98 | 7.5 | 92.5 |
| Weighted KNN | 0.11 | 0.79 | 0.86 | 20 | 80 |
| Subspace discriminant | 0.13 | 0.71 | 0.87 | 28.1 | 71.9 |
| Subspace KNN | 0.08 | 0.47 | 0.77 | 35.6 | 64.4 |

**Table 7.** Confusion matrix of diseased vs. healthy mango leaves.

| Classification Class | Classification Class | | |
|---|---|---|---|
| | **Healthy** | **Powdery Mildew** | **Sooty Mold** |
| Healthy | 97.8% | <1% | - |
| Powdery mildew | - | 97.8% | <1% |
| Sooty mold | - | - | 91.1% |

*4.4. Discussion*

The achieved results are more efficient than the results presented by Srunitha et al. [32] in 2018 who introduced k-means for the segmentation of the diseased part and a multiclass SVM for classification purposes and obtained an accuracy of 96%, as shown in Table 8. However, the list of diseases they detected did not contain powdery mildew, whereas our proposed method showed an accuracy of 95.5% while detecting powdery mildew as well. None of the studies available have yet checked the vein pattern of plant leaves. As it is the most important part of the plant as concerns food and water transportation, the proposed technique is far superior to the available techniques. Furthermore, the mobile-based system proposed by Anantrasirichai et al. [38] in 2019 uses classification techniques and obtained an accuracy of 80% when detecting the diseases of mango plants. In comparison, the achieved accuracy of the proposed model is much better, at 95.5%.

**Table 8.** Comparison of different segmentation and classification techniques with proposed model.

| Methods | Year | Technique | Accuracy (%) |
|---|---|---|---|
| K means by Srunitha et al. [32] | 2018 | Multiclass SVM | 96% |
| Mobile phone based by Anantrasirichai et al. [38] | 2019 | Classification | 80% |
| Proposed | | Leaf vein-seg | 95.5% |

## 5. Conclusions

A novel segmentation technique was introduced in this paper. Two types of mango leaf disease, powdery mildew and sooty mold, were recognized. A self-collected dataset was used to perform this task. A leaf vein segmentation technique that detects the vein pattern of the leaf was proposed. The leaf's features were extracted on the basis of color and texture after performing the segmentation. Ten different classifiers were used to obtain the results. The overall performance of the proposed method is much improved compared to already available methods. However, the following improvements will be considered in the future: (1) increase the number of images in the dataset, (2) minimize the identification time through feature optimization algorithms [39–41] to implement it in real time, and (3) implement some latest deep learning models [42–45].

**Author Contributions:** Conceptualization, R.S. and J.H.S.; methodology, R.S., J.H.S., and M.S.; software, R.S. and M.S.; validation, J.H.S., M.Y., and M.S.; formal analysis, M.Y. and J.C.; investigation, H.-S.Y. and J.C.; resources, M.Y. and J.C.; data curation, H.-S.Y. and J.C.; writing—original draft preparation, R.S. and M.S.; writing—review and editing, J.C. and M.Y.; visualization, M.S. and M.Y.; supervision, J.H.S. and J.C.; project administration, H.-S.Y.; funding acquisition, H.-S.Y. All authors have read and agreed to the published version of the manuscript.

**Funding:** This work was supported by the National Research Foundation of Korea (NRF) grant funded by the Korea government (Ministry of Science and ICT; MSIT) under Grant RF-2018R1A5A7059549.

**Institutional Review Board Statement:** Not applicable.

**Informed Consent Statement:** Not applicable.

**Conflicts of Interest:** The authors declare no conflict of interest.

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
