# Peer review of "Mango Leaf Disease Recognition and Classification Using Novel Segmentation and Vein Pattern Technique"

_applsci, doi:10.3390/app112411901_

Round 1

Reviewer 1 Report

This manuscript reports a study, that combines image processing technology and multiple machine learning methods, to identify diseased mango plant leaves. Plant disease recognition based on image features is a research hotspot of plant informatics. It is conducive to help agricultural countries recover losses, and has high application value. However, there are some problems with the manuscript and major revisions are required.

  1. There are some grammatical errors in this article that need to be corrected.
  2. The abstract needs to be modified. It is a brief statement of the main content of the paper,which is no need to describe too much background.

  3. Line 113, you think there are some weaknesses in the existing literature , and you have improved them in your manuscript. However, I did not see the weaknesses you described in the above-mentioned literature.

  4. Line 89-90,references cited: C. Pane et al. [16] adopted a ML technique with the wavelength between 403 to 446nm. I don't know what it means that ML has a wavelength of 403 to 446 nanometers. What kind of experimental results have been achieved, and how are this technology applied to plant disease detection?
  5. table 8,I can't find literature [30] in the literature list.

  6. The biggest problem is that I think a simple transformation of the data image, like translation and flipping, is meaningless to expand the database.

    • The texture features extracted using LBP are rotation invariant, which means the feature will not change due to the rotation of the image or the subject. So can the rotated data really achieve expansion?
    • Rotation and flipping doesn't change the color of the image, which is not very helpful for the extraction of color features.

    So I think it is necessary to expand the laboratory data in a meaningful way.

  7. What are the traditional segmentation methods for detecting plant leaf diseases? The research has been improved and innovated. Whether the technology you used improves the experimental results should be compared and explained.

  8. In the discussion section, you only compared the recognition accuracy  with other experiments. The content should be expanded.

Author Response

Response sheet attached. thank you

Reviewer 2 Report

  • Abstract: Authors should re-write the abstract to make more concise and highlight the recommendations from the use of the “segmentation approach”. “Man is a classic fruit and in great demand” – The sentence is not clear. What does authors meant by “Mango is classic fruit”. Additional information on the results from the study should be included.
  • Introduction: There are a lot inconsistent sentence structure in the section. It has to re-written. Authors should consider the following in the introduction:

(1) The justification of the study should be further strengthened.

(2) The gap in the literature should be identified and the contribution of the study should be clearly stated and expanded.

  • Literature: This section as it stands needs improvements. The studies cited are so disjointed and not well integrated. Authors should consider improving the flow from one paragraph to the other.

Line 120 – 122: The purpose of this paragraph is not clear. Authors should consider expanding this, or otherwise remove.

  • Units of the data in Table 2 should stated
  • The discussion of the results in Section 4.4 should be expended.
  • Conclusion: Authors should consider improving this section to include more information on the implication of the novel approach developed in this study.
  • On the overall, the paper has a merit for publication in this journal but needs to be improved.

Author Response

response sheet attached. thank you

Round 2

Reviewer 1 Report

After revision, this article meets the requirements for publication.

Author Response

Response Sheet. 

Reviewer 2 Report

The authors have diligently responded to all the comments that I raised on the earlier draft. The paper is now fit for publication.

Author Response

Response sheet has been attached. thank you
